
# Air Density Induced Error on Wind Energy Estimation

Aurore Dupré[1], Philippe Drobinski[1], Jordi Badosa[1], Christian Briard[2], and Riwal Plougonven[1]

[1]LMD/IPSL, École Polytechnique, Institut Polytechnique de Paris, ENS, PSL, Research University, Sorbonne Université, CNRS, Palaiseau, France
[2]Zephyr ENR, Saint-Avertin, France

**Correspondence:** Aurore Dupré (aurore.dupre@lmd.polytechnique.fr)

**Abstract.** In recent years, environmental concerns have encouraged the use of wind power as a renewable energy resource. However, high penetration of the wind power in the electricity system is a challenge due to the uncertainty of wind energy forecast. Estimation of the wind enregy production requires a forecast for the wind (the main source of uncertainty) but also of density, often overlooked. Measure of air density is a key for more accurate wind energy prediction. Wind farms often lack

instrumentations of temperature and pressure, needed for accurate air density estimation at hub height to be used for locally debiasing air density forecast. In this study, the error budget of air density estimate is computed distinguishing temperature and pressure contributions. The analysis uses measurements for in-depth local analysis as well as meteorological reanalysis to investigate the added-value of a model-based value when measurement is missing. Meteorological reanalysis is also used to study spatial pattern of error budgets (mountainous area, coastal regions, plains, ...). The effect of altitude is carefully

accounted for. Temperature is by far the variable inducing the largest errors when it is missing in the air density correction, and replaced by the standard atmosphere value (i.e. 15°C, used as reference in power curves). It is particularly true for very cold or warm conditions (i.e. far from the standard value), for which the error on wind energy production is nearly halved when an accurate correction of temperature is performed.

## 1   Introduction

Over the past two decades, the global energy market is turning increasingly to green energies. In this context, the wind energy sector has soared all over the world. Wind farms are located in more than 90 countries around the world, 9 of them with an installed capacity of more than 10 GW, and 30 with more than 1 GW across Europe, Asia, North America, Latin America and Africa. In 2017, 52.5 GW of new wind power was installed across the globe, bringing total installed capacity up to 539 GW. In France, wind power installation increased by 14.04% in 2017 (Fried et al., 2017), especially thanks to the feed-in tariffs.

However, the variable and intermittent nature of the wind energy source limits its high penetration and wind farm operators need to be able to predict their production as accurately as possible to avoid to pay fines. In order to optimize the wind energy production, with respect to the bid, it is essential to reduce the error related to wind energy forecasts.

Most of the time, wind energy is computed from the wind speed through a power curve. The theoretical power curve is provided by the wind turbine manufacturer for standard temperature ($T_0 = 15$°C) and pressure ($P_0 = 1013.25$ hPa). This curve

gives the wind energy production as a function of the wind speed. Deriving empirical power curve for a given turbine is a key



for more accurate wind power estimate. This is still a very active research field (Lydia et al., 2014; Carrillo et al., 2013). Studies have shown large sensitivity to the empirical estimation method of the of wind power estimate with errors ranging reaching 50% (Lydia et al., 2013), and varying by about 20% between parametric and non-parametric approaches (Shokrzadeh et al., 2014).

However, the power performance of a wind turbine also depends on air density. But most studies neglect it (Wagenaar and Eecen, 2011). Its impact is not negligible with an error on wind power estimate which can be reduced by 20% when temperature correction for air density is accounted for (Fischer et al., 2017). However as for the power curve, the sensitivity to the methods used to correct for air density is extremely large with errors varying by more than 100% depending on the method (Pelletier et al., 2016).

Accurate estimate of air density is therefore a key to reduce the uncertainty of the wind energy production forecast. In an operational configuration, different strategies can be adopted to achieve this. Considering default values is clearly the worst strategy and it is equivalent to ignoring air density variations. The best strategy requires real time temperature and pressure measurements for an a priori empirical derivation of the power curves and an a posteriori method for debiasing locally wind energy production forecasts. However, wind farms equipped with both sensors are rare. In that case, values from Numerical
Weather Prediction models may be a suitable alternative.

This paper aims at underscoring the temperature and pressure contributions of the air density computation in order to better take into account the lack in wind farm instrumentation. Section 2 details the methodology to compute the error budget of the air density with an in-depth analysis of the temperature and pressure contributions. The error budget analysis is performed at a densely instrumented site and its spatial pattern and sensitvity to the terrain complexity is further investigated using
meteorological analysis. Application to wind energy production estimation is shown at an actual wind farm in Section 3. Section 4 concludes the study.

## 2   Air density error budget

### 2.1   At the SIRTA observatory

To quantify the contributions of temperature and pressure in the air density error budget, we use the large observation dataset
from the SIRTA observatory (Site Instrumental de Recherche par Télédétection Atmosphérique), located 20 km South of Paris (France) (48.7°N, 2.2°E, 150 m altitude) (Haeffelin et al., 2005). We retrieve surface pressure and temperature at 2 m at 10-minutes frequency from 2015 to 2017 to compute the air density and the different contributions. We compute the air density $\rho$ from the temperature $T$ and pressure $P$ based on the ideal gas law $P = \rho \frac{R}{M} T$ as:

$$\rho = \frac{MP}{RT} = \frac{M}{R} \frac{(P_0 + P')}{(T_0 + T')} = \underbrace{\frac{MP_0}{RT_0}}_{\rho_0} \Big(1 + \frac{P'}{P_0}\Big) \Big(\frac{1}{1 + T'/T_0}\Big) \tag{1}$$

where $P_0 = 1013.25$ hPa and $T_0 = 288.15$ K are reference values of the standard atmosphere at the Earth's surface. The quantitites $P'$ and $T'$ are the deviations to the reference values, $M = 0.02898$ kg mol$^{-1}$ is the dry air molar mass and $R =$





8.31 J K$^{-1}$ mol$^{-1}$ is the ideal gas constant. To quantify the contributions of the temperature and pressure to the air density error budget, we compute the normalized bias (BIAS), the normalized mean absolute error (MAE) and the normalized root mean square error (NRMSE) as:

$$\text{BIAS} := \frac{1}{\bar{y}}\left(\frac{1}{n}\sum_{i=1}^{n}\left(y_i - \hat{y}_i\right)\right) \tag{2a}$$

$$\text{MAE} := \frac{1}{\bar{y}}\left(\frac{1}{n}\sum_{i=1}^{n}|y_i - \hat{y}_i|\right) \tag{2b}$$

$$\text{NRMSE} := \frac{1}{\bar{y}}\sqrt{\frac{1}{n}\sum_{i=1}^{n}(y_i - \hat{y}_i)^2} \tag{2c}$$

where, $y_i$ is a measured variable (air density), $\hat{y}_i$ is the computed variable, $n$ is the sample size and $\bar{y}$ is the mean value over the period ranging between 2015 and 2017.

Table 1 displays the values of BIAS, MAE and NRMSE between the measured and computed air density. Two ways of computing it are assessed. The first row corresponds to the values when the pressure is set to its reference value ($P = P_0$), only temperature deviation is considered (hereafter referred as "temperature contribution"). The second row corresponds to the values when the temperature is set to its reference value ($T = T_0$), only pressure deviation is considered (hereafter referred as "pressure contribution"). The two contributions are evaluated separately because a wind farm may only have access to pressure or temperature measurements. When $P = P_0$ (temperature contribution, upper row), the BIAS and MAE have very similar absolute value (1.34 % and 1.37 % with respect to the reference density), suggesting a significant negative bias.The NRMSE is of the same order of magnitude. When $T = T_0$ (pressure contribution, lower row), the bias is positive (+1.22 % with respect to the reference density). The MAE and NRMSE are more than 1.5 times larger than when $P = P_0$, suggesting that the temperature contribution has a larger weight in the air density error budget. Indeed, despite the averaged relative fluctuations $\frac{P'_{\text{OBS}}}{P_0}$ and $\frac{T'_{\text{OBS}}}{T_0}$ have the same order of magnitude (around 3 K over 300 K for the temperature and 10 hPa over 1000 hPa for the pressure), the relative standard deviation $\frac{\sigma_P}{P_0}$ and $\frac{\sigma_T}{T_0}$ is around $2.42 \times 10^{-2}$ and $8.66 \times 10^{-3}$ respectively. The larger temperature variability causes a larger impact of temperature on air density error budget.

However, wind farm operators often lack simultaneous real time temperature and pressure measurements at hub height to compute air density and correct accordingly the wind energy production. Meteorological reanalysis, analyses or even short term forecasts are supposed to be the best 3 D representation of the state of the atmosphere at a given time. We use here the temperature and pressure at 2 m from ERA5 reanalysis to test the added value of NWP model output when local measurements are missing. ERA5 are reanalysis dataset provided by the European Center for Medium-Range Weather Forecasts (ECMWF). ERA5 provides hourly estimates of a large number of atmospheric, land and oceanic climate variables. The data cover the Earth on a 30 km grid and resolve the atmosphere using 137 levels from the surface up to a height of 80 km. The grid point nearest to SIRTA (48.75°N, 2.25°E) is located less than 7 km away. In order to have the same time resolution we compute hourly averaged of the 10-minutes measurements. Table 2 displays the error indicators BIAS, MAE and NRMSE computed by comparing the air density measured at the SIRTA observatory with that estimated with the temperature and pressure from





|  | BIAS (in %) | MAE (in %) | NRMSE (in %) |
|---|---|---|---|
| $\tilde{\rho} = \rho_0 \left( \dfrac{1}{1 + T'_{\mathrm{OBS}}/T_0} \right)$ | -1.34 | 1.37 | 1.61 |
| $\tilde{\rho} = \rho_0 \left( 1 + \dfrac{P'_{\mathrm{OBS}}}{P_0} \right)$ | 1.22 | 2.22 | 2.72 |

**Table 1.** Bias (BIAS), mean absolute error (MAE) and normalized root mean square error (NRMSE) for air density when the pressure is set to a reference value ($P = P_0 = 1013.25$ hPa) ("temperature contribution, upper row) and when the temperature is set to a reference value ($T = T_0 = 288.15$ K) ("pressure contribution, lower row). The data used to compute the error indicators are measurements collected at SIRTA observatory, located 20 km South of Paris (France) (48.7°N, 2.2°E, 150 m altitude).

ERA5 reanalysis. The results are very comparable to those of Table 1 (see middle and lower rows). Surprisingly, the errors with ERA5 are slightly lower than the errors with the measurements. This can be explained by the chosen reference values which minimize the deviations in case of the reanalysis.

|  | BIAS (in %) | MAE (in %) | NRMSE (in %) |
|---|---|---|---|
| $\tilde{\rho} = \rho_0 \left( \dfrac{1}{1 + T'_{\mathrm{NWP}}/T_0} \right) \left( 1 + \dfrac{P'_{\mathrm{NWP}}}{P_0} \right)$ | -0.44 | 0.47 | 0.58 |
| $\tilde{\rho} = \rho_0 \left( \dfrac{1}{1 + T'_{\mathrm{NWP}}/T_0} \right)$ | -1.29 | 1.33 | 1.57 |
| $\tilde{\rho} = \rho_0 \left( 1 + \dfrac{P'_{\mathrm{NWP}}}{P_0} \right)$ | 0.74 | 2.06 | 2.53 |

**Table 2.** Same as Table 1 with data from ERA5 reanalysis at the grid point nearest to SIRTA observatory. The additional upper row compares the air density computed from ERA5 data with the measured air density.


In case only one variable is measured (temperature or pressure), Table 3 displays the BIAS, MAE and NRMSE using ERA5 for the missing variable. For instance, if the temperature is measured ($T_{\mathrm{OBS}}$) and not the pressure, then the pressure from the model output ($P_{\mathrm{NWP}}$) is used. Conversely, if the pressure is measured ($P_{\mathrm{OBS}}$) and not the temperature, then the temperature from the model output ($T_{\mathrm{NWP}}$) is used. All error indicators (BIAS, MAE and NRMSE) are lower compared to those computed by

discarding the missing variable, should it be measured (Table 1) or obtained from model output (Table 2).

## 2.2   Spatial pattern

In this part, the impact of both contributions is investigated across France. In both cases, the reference air density is computed from model outputs. Figure 1 displays the NRMSE of pressure contribution (left column) and temperature contribution (right column) over France. All data are retrieved from ERA5 reanalysis. The errors are low for the most part of France ($< 4$ % when $T = T_0$ and $< 5$ % when $P = P_0$) except in the mountains (up to 6 % when $T = T_0$ and up to 30 % when $P = P_0$). This is due






|  | BIAS | MAE | NRMSE |
| --- | --- | --- | --- |
|  | (in %) | (in %) | (in %) |
| $\tilde{\rho} = \rho_0 \left( \dfrac{1}{1 + T'_{\mathrm{OBS}}/T_0} \right) \left( 1 + \dfrac{P'_{\mathrm{NWP}}}{P_0} \right)$ | -0.49 | 0.49 | 0.50 |
| $\tilde{\rho} = \rho_0 \left( \dfrac{1}{1 + T'_{\mathrm{NWP}}/T_0} \right) \left( 1 + \dfrac{P'_{\mathrm{OBS}}}{P_0} \right)$ | 0.05 | 0.27 | 0.36 |

**Table 3.** Same as Table 1 when measurements at SIRTA observatory and data from ERA5 reanalysis at the grid point nearest to SIRTA observatory are combined to compute the air density.

to the reference values $P_0 = 1013.25$ hPa and $T_0 = 288.15$ K. These are approximations which are not valid anymore at high altitudes.

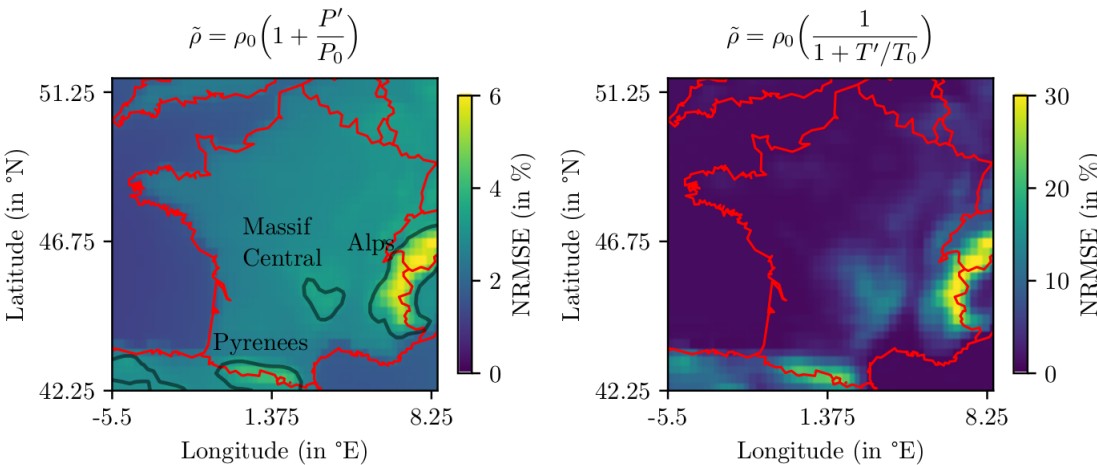

**Figure 1.** The left column displays the error when $T = T_0$ and the right column displays the error when $P = P_0$. Both figures display the NRMSE in %. The data are retrieved from ERA5 reanalysis.

To overcome this problem, we compute the reference temperature and the pressure corrected with the altitude according to the
International Standard Atmosphere (ISA) as (ISA, 1975) :

$$\tilde{P}_0 = P_0 \left( 1 - \frac{0.0065}{T_0} z \right)^{5.255} \tag{3a}$$

$$\tilde{T}_0 = T_0 - \frac{6.5}{1000} z \tag{3b}$$

with $z$, the altitude in meters.





After correction, the errors do not exceed 1 % except in the Alps where the NRMSE is around 3.0 % when $T = \tilde{T}_0$ and
1.2 % when $P = \tilde{P}_0$ (figure not shown), while it was around 6.0 % when $T = T_0$ and around 30 % when $P = P_0$ (see Fig. 1).
Considering a constant value of temperature introduces larger errors than considering a constant value of pressure. Again, this
is due to the higher variability of the temperature.

## 3  Application to a wind farm

We consider the data of a wind farm located in Bonneval, a small town 100 km Southwest of Paris (48.20°N, 1.42°E, 135 m
altitude). The wind farm is operated by Zephyr ENR, a private company managing 5 other wind farms. The Bonneval wind
farm called "Parc de Bonneval", has been set up in 2006 and has 6 Vestas V80-2 MW turbines of 100 m height. In "Parc de
Bonneval", anemometers and thermometers are installed on each turbine. The data are 10-minutes averaged. For consistency,
the dataset analyzed here has been collected between 2015 and 2017. The power curve is retrieved by averaging the wind speed
and power measurements at the 6 turbines. The wind speed measurements are binned into 0.5 m s$^{-1}$ intervals. The wind energy
production is averaged in each bin and the power curve is retrieved by fitting the mean wind energy production as function of
the mean wind speed.

According to (IEC, 2005), to take into account the air density, the wind speed must be normalized as follows:

$$U_n = U_t \left( \frac{\rho_t}{\rho_0} \right)^{1/3} \tag{4}$$

where $\rho_0 = 1.225$ kg m$^{-3}$ is the standard density for which the power curve is given by the wind turbine manufacturer.
Applying Eq. 4 is an efficient way to account for the air density. It is less costly than training a dedicated model and it has
already been used for energy potential evaluation (Dhamouni et al., 2011). As the power curve is given as a function of
the wind speed only and a reference density, it is necessayr to incorporate the density variations in the value of the wind.
Figure 2 displays different power curves fitted for several temperature intervals. In Fig. 2a, the wind speed is directly taken
from the measurements: $U_n = U_t$. For a wind speed of 6.5 m s$^{-1}$, the energy production varies from 356 kW for a temperature
between 0°C and 5°C to 298 kW for a temperature between 30°C and 35°C (i.e. 19.5 % difference). Figure 2b shows how the
normalization given by Eq. (4) corrects the effect of the air density variations. The temperature is retrieved from measurements
and the pressure from ERA5 reanalysis nearest grid point (48.25°N, 1.5°E) around 8 km from "Parc de Bonneval". For a wind
speed of 6.5 m s$^{-1}$, the energy production varies after air density correction from 346 kW for a temperature between 0°C and
5°C to 320 kW for a temperature between 30°C and 35°C (i.e. 8.1 % difference). The spread between the power curves is
therefore highly reduced.

Table 4 displays BIAS, MAE and NRMSE between the measured and modeled wind energy production for the measured
wind speed ($U_n = U_t$) and normalized wind speed using Eq. (4). The normalization is here applied with respect to the nominal
power, equal to 2 MW. One can first note that the bias is close to 0. The negligible bias can be explained by the fact that on



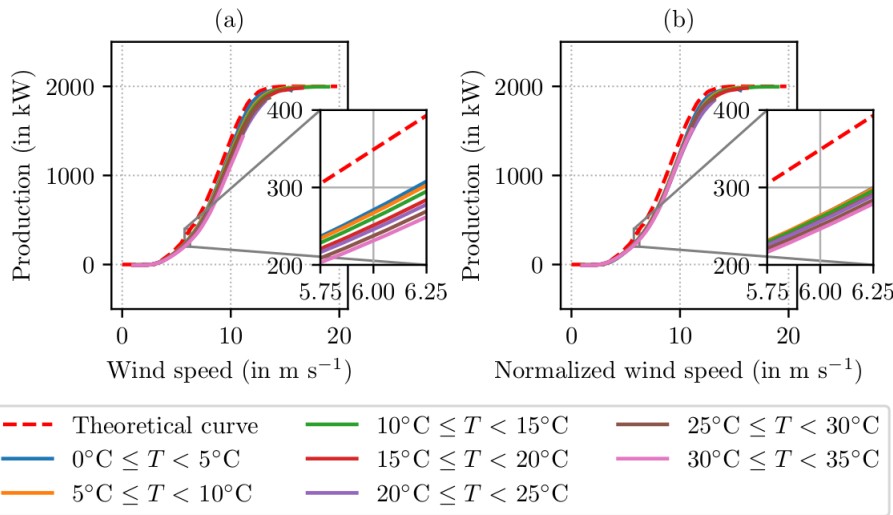

**Figure 2.** Power curves averaged over the 6 wind turbines of "Parc de Bonneval' as a function of temperature ranges, when the wind speed is not corrected for air density variation ($U_n = U_t$) (a) and when it is corrected following Eq. (4) (b). The theoretical power curve provided by the manufacturer is shown in dashed red.

average at this location, the temperature and pressure conditions are close to the reference values. Correcting for air density variation reduces MAE and NRMSE, as they are indicators quantifying the spread which is reduced (Fig. 2). However, the improvement is significant but remains low as MAE goes from 0.96 % (no normalization) to 0.77 % (normalization), and NRMSE from 1.58 % (no normalization) to 1.32 % (normalization). As explained earlier, the temperature and pressure conditions are close to the reference values (i.e. the mean temperature for this period is around 13°C and the mean pressure is around 1000 hPa) so the averaged improvement is weak (Ahmed Shata and Hanitsch, 2006).

|  | BIAS | MAE | NRMSE |
|---|---|---|---|
|  | (in %) | (in %) | (in %) |
| No normalization : $U_n = U_t$ | 0.03 | 0.96 | 1.58 |
| $U_n = U_t \left( \left( 1 + \dfrac{P'}{P_0} \right) \left( \dfrac{1}{1 + T'/T_0} \right) \right)^{1/3}$ | 0.02 | 0.77 | 1.32 |

**Table 4.** BIAS, MAE and NRMSE between the measured and modeled wind energy production for the measured wind speed ($U_n = U_t$) and normalized wind speed using Eq. (4).


Focusing on more extreme conditions such as temperatures below 5°C or higher than 25°C, improves significantly the impact of the air density correction. Table 5 summarizes these results. Compared to Table 4, the differences between the normalized and measured wind speeds are larger. For instance, the MAE improves by about 20 % for all cases (Table 4) to about 33 %





for extreme cases only (Table 5, $T \geq 25$°C). Similarly, NRMSE improves by about 17 % in all cases (Table 4) to about 37 %
for extreme cases only (Table 5, $T \geq 25$°C). Those extreme events are not so rare because cold temperatures lower than 5°C
(mainly winter months) occur 10.7 % of the time and hot temperatures higher than 25°C occur (mainly summer months) 5.5 %
of the time.

| | | BIAS (in %) | MAE (in %) | NRMSE (in %) |
|---|---|---|---|---|
| $T \leq 5$°C | $U_n = U_t$ | 20.55 | 27.66 | 2.12 |
| | $U_n = U_t\left(\left(1+\dfrac{P'}{P_0}\right)\left(\dfrac{1}{1+T'/T_0}\right)\right)^{1/3}$ | 6.19 | 18.36 | 1.55 |
| $T \geq 25$°C | $U_n = U_t$ | -19.51 | 24.09 | 1.89 |
| | $U_n = U_t\left(\left(1+\dfrac{P'}{P_0}\right)\left(\dfrac{1}{1+T'/T_0}\right)\right)^{1/3}$ | -5.23 | 14.51 | 1.19 |

**Table 5.** BIAS, MAE and NRMSE between the measured and modeled wind energy production for temperatures lower than 5°C and higher than 25°C. Comparison between measured wind speed ($U_n = U_t$) and normalized wind speed (normalization using Eq. (4)) is shown.

## 4  Conclusions

This paper assesses the adding value of the wind normalization to take into account the air density in the wind energy modeling
using in-situ measurements and meteorological analysis. In the state of the art, most of the papers that take into account
the air density use non-parametric methods. Those methods are numerically costly. Our parametric method overcomes this
issue with also significant improved results with respect to results which do not account for air density correction. Indeed,
this study shows that a correction for air density improves the wind energy production estimation by more than 15% over
the 3 investigated years (2015 to 2017). In most of the paper that deal with this normalization, there is no skill scores of the
improvements due to the normalization but for instance, visual comparison between power curves. Moreover, when skill scores
are given, they are given without distinction of the atmospheric conditions and without comparison with the case for which the
air density is not considered. The lack of interest for this issue lies in the fact that the overall improvement remains limited
especially in mid-latitudes, where atmospheric conditions are close to the standards. In this study, the usefulness of the air
density correction is highlights by enhancing the situations where atmospheric conditions are far from the standard conditions
and the improvement reaches nearly 40% in those cases (temperatures below 5°C or above 25°C). This study also shows that
the temperature is the key variable to account for when correcting for air density as its impact is the largest on the uncertainty
of the air density estimation (twice larger than the pressure term). Meteorological analysis (i.e. model-based observations)
have also a beneficial impact when one of the key variable (temperature or pressure) or even both variables are not measured.





Correction for altitude using standard atmosphere is the simplest and most efficient way to correct for air density when finer

information is not available.

*Acknowledgements.*  The authors would like to thank the French company Zephyr ENR for providing the data of the "Parc de Bonneval" and for supporting this research. This work also contributes to TREND-X program on energy transition at École Polytechnique.





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
