# Peer review of "Air Density Induced Error on Wind Energy Estimation"

_Annales Geophysicae, 2019_

## Referee Comment (RC1) · Anonymous Referee #1 · 28 Jul 2019

The article deals with several aspects of air density calculation with respect to wind power production. The first part of the article shows, in one location, the difference between air density calculated from standard values of air temperature and pressure (15°C, 1013.25 hPa) and from their site-specific values (measured and estimated from reanalysis) in an hourly time series. The second part provides a brief look how the errors deviate over France - this is mostly because of systematic decrease of air pressure and temperature with altitude. The third part shows the impact of air temperature and pressure variability on energy production at specific wind farm.

In common practice it is well known and understood that air density is an important factor that affects the power production of wind turbines. It is a standard in wind energy industry that the wind turbine production is corrected for air density effects where it matters. It is usual that the data from reanalyses or measurements (for long-term prediction of energy yield) or from operational NWP models (for short-term forecasts)

[Figure]

are applied.

From this perspective, the content of the article seems to be behind the common practice in wind industry. It mostly consists of trivial comparisons between common data sources. It explores the error that occurs when standard air pressure and temperature is used – but the standard air density is in practical application not used if the error matters. Also the dependence of power production on air density is well known, at least at such a high level as it is presented in this article. As far as I understood the article, it neither provides in depth exploration or explanation of a relevant issue in wind energy practice, nor does it bring a new finding for general knowledge. For this reason, if the authors would not explain that I missed some important contribution of the article, I do not recommend it for publication in scientific journal. Perhaps it would make sense to concentrate on some more specific issue and explore it in more depth and in a more comprehensive way.

As my recommendation is to reject the article, I did not formulate more specific comments. I can add them in case the article was still admitted for publication. Generally, in my opinion there are several results and topics touched by the article that would deserve more detailed and thoughtful explanation or interpretation. For example, it would be worth a comment why the NRMSE is much lower than MAE and BIAS in Table 5.

---

## Referee Comment (RC2) · Anonymous Referee #2 · 29 Jul 2019

A. Air density affects the wind turbines power curve accuracy. IEC standard, WT power curves are calculated using a data reduction technique known as binning using IEC standard in which Air density correction is carried out before binning. Similar research on air density impact on power curve is already being carried out and published in the following articles 1. Pandit, RK, Infield, D, Carroll, J. Incorporating air density into a Gaussian process wind turbine power curve model for improving fitting accuracy. Wind Energy. 2019; 22: 302– 315. https://doi.org/10.1002/we.2285.

2. Bulaevskaya V, Wharton S, Clifton A, Qualley G, Miller WO. Wind power curve modeling in simple and complex terrain using statistical models. J. Renewable Sustainable Energy. 2015;7(1):013103. https://doi.org/10.1063/1.4904430

So novelty is doubtful.

B. As per the IEC standard, the air density correction shall be applied when the site

density differs from the standard value (1.225 kg/m3) by more than 0.05 kg/m3. Does your data include this in your research? and how it affects the power curve accuracy?

C. The author ignores the latest work carried out in the area of air density and power curve. Hence, work seems to be behind.

Therefore, based on the above comments, my recommendation is to reject the paper.
* * *

---

## Referee Comment (RC3) · Anonymous Referee #3 · 31 Jul 2019

I cannot see the value of this paper. The central equation applied to air density correction is taken from the IEC Standard and is widely applied by the industry. Despite this a number of papers have been published recently (eg Pandit, RK, Infield, D, Carroll, J. Incorporating air density into aGaussian process wind turbine power curve model for improving fitting accuracy. WindEnergy. 2019; 22: 302– 315.) that demonstrate more accurate approaches. These should not be dismissed as in this paper on the basis that they require a degree of computation.

The density correction equation (4 in the paper) is only to be applied below rated paper for pitch controlled turbines (the norm these days). The equation is an approximation and a detailed study of the errors involved in its application across a range of wind farms exposed to widely differing air density would be of research interest.

One minor comment: the first part of the paper compares computed air density with

measured values. But what are these measured values; and how are they obtained since direct measurement of air density is difficult to undertake.

---

## Author Comment (AC1) · 8 Aug 2019

First of all thank you for your comment. We should have specify the context more clearly. This study targets small structures like wind energy producers with small wind farms. Typically, this work was conducted thanks to a wind energy producer called Zephyr ENR who owns 6 farms with six wind turbines each. In this context, the accuracy of the forecasts is crucial.

According to IEC, the density normalization should be applied when the air density differs by more than 0.05 kg m$^{-3}$ from the standard air density. This is not the case for the considered wind farm. However, we have shown that even if the variation in the air density seems weak over the entire year, during summer or winter taking into account those variations leads to an improvement of nearly 40% in the wind energy modeling. Then, this should not be neglected.

[Figure]

Using the standard temperature and pressure in the formula means not taking them into account. Consequently we do not explore the error that occurs when standard air pressure and temperature are used but we explore the error when pressure and temperature are neglected. Then, we compare those cases with the cases where they are considered. Indeed, small companies cannot necessarily afford to pay for analyses, hence the study on the neglect of the pressure or temperature term.

In this study, the impact of density variations on the wind power modeling is accurately quantified using in-situ data over 3 years. The use of in-situ data, especially over such a long period of time, is not systematic in the literature and that is one of the strengths of this study. We use the simplest method in order to demonstrate that even this naive approach allows significant improvements. The literature review presents the other methods and their associated improvements. It has been expanded thanks to the referee's comment.

Finally, thank you for noticing the highest values of MAE and BIAS in Table 5. Those values were in kW instead of %. This has been corrected.

---

## Author Comment (AC2) · 8 Aug 2019

Thank you for your comment.

**A.** Indeed several researches deal with the impact of air density on the power curve. The literature review has been expanded following the referee's comment and we now detail some of these methods and their performances. In this work, we use the simplest approach in order to demonstrate that even this naive method allows significant improvements and even if the air density variations are weak, the impact remains significant. Of course, some sophisticated methods also lead to significant improvements and even if the comparison with the IEC normalization is not systematic (making comparisons difficult) they are now presented with their associated improvements. The purpose of this work is to highlight the impact of air density (even when variations are statistically low) on power modeling using measurements directly collected on site.

[Figure]

**B.** To briefly recall the context, this study targets small structures like wind energy producers with small wind farms. For instance, this work was conducted thanks to a wind energy producer called Zephyr ENR who owns 6 farms with six wind turbines each. In this context, the forecasts accuracy is crucial. Indeed, according to IEC, the density normalization should be applied when the air density differs by more than $0.05$ kg m$^{-3}$ from the standard air density. This is not the case for the entire period (3 years) at the considered wind farm. However, we have shown that even if the variation in the air density seems weak over the entire year, it may lead to improvements up to 40% in the wind energy modeling. Consequently, this induced significant errors that should not be neglected. With regard to the power curves, Fig 2 shows the impact on their accuracy with a variation up to 20% in the power output depending on the air density.

**C.** The strength of this work is the use of in-situ data which is not systematic in the literature even in the latest study and especially for such a long time period. Nevertheless, the literature review has been improved by including the latest papers as suggested by the referee.

---

## Author Comment (AC3) · 8 Aug 2019

First of all, the authors would like to thank the referee for his comment.

This study targets small structure like wind energy producers with small wind farms. Typically, this work was conducted thanks to a wind energy producer who owns 6 farms with six wind turbines each. Then, the forecast accuracy is vital. In this context, the impact of density variations on the wind power modeling is accurately quantified using in-situ measurements over 3 years. The use of in-situ data, especially over such a long period of time, is rare in the literature and that is one of the strengths of this work. Indeed this work focus on the impact at one wind farm. But the second part, about the spatial study and the part on power output modeling can be combined to get an idea of the spatial impact. Unfortunately, we do not have access to wind farms data exposed to widely differing air density. But the period of time over which this study is conducted

is sufficient to guarantee the robustness of this work.

We use the simplest method in order to demonstrate that even this naive approach allows significant improvements (up to 40%). The literature review presents the other methods and their associated improvements even if the comparison with the IEC normalization is not systematic (making comparisons difficult). In any case, the literature review has been expanded thanks to the referee's comments.

In the first part of the paper we use in-situ measurements of pressure and temperature to compute the air density through the following formula: $\frac{MP}{RT}$ where $M$ is the molar mass of dry air and $R$ is the ideal gas constant.
* * *